# Exploring HIV Vaccine Progress in the Pre-Clinical and Clinical Setting: From History to Future Prospects

**DOI:** 10.3390/v16030368

**Published:** 2024-02-27

**Authors:** Amitinder Kaur, Monica Vaccari

**Affiliations:** 1Division of Immunology, Tulane National Primate Research Center, Covington, LA 70433, USA; akaur@tulane.edu; 2School of Medicine, Tulane University, New Orleans, LA 70112, USA

**Keywords:** HIV epidemic, vaccine development, non-human primate model (NHP)

## Abstract

The human immunodeficiency virus (HIV) continues to pose a significant global health challenge, with millions of people affected and new cases emerging each year. While various treatment and prevention methods exist, including antiretroviral therapy and non-vaccine approaches, developing an effective vaccine remains the most crucial and cost-effective solution to combating the HIV epidemic. Despite significant advancements in HIV research, the HIV vaccine field has faced numerous challenges, and only one clinical trial has demonstrated a modest level of efficacy. This review delves into the history of HIV vaccines and the current efforts in HIV prevention, emphasizing pre-clinical vaccine development using the non-human primate model (NHP) of HIV infection. NHP models offer valuable insights into potential preventive strategies for combating HIV, and they play a vital role in informing and guiding the development of novel vaccine candidates before they can proceed to human clinical trials.

## 1. Introduction

Human immunodeficiency virus (HIV) continues to impose a significant global health impact, affecting millions of individuals annually with new infections. Despite progress in treatment and prevention approaches, the need for an effective HIV vaccine remains urgent. Vaccines have historically played a crucial role in controlling and eradicating infectious diseases, and an effective HIV vaccine could be transformative in curtailing the epidemic. Developing a preventive vaccine for HIV poses unique challenges due to the virus’s ability to target the immune system, mutate rapidly, establish latent reservoirs, and evade immune responses. Substantial research efforts have been dedicated to understanding the complex biology of HIV and developing safe and effective vaccine candidates. The evolution of HIV vaccine research spans almost four decades and aligns with an enhanced comprehension of HIV and host immune responses.

***History of HIV Vaccine Development*.** The history of HIV vaccine development encompasses almost four decades, with initial emphasis placed on the role of antibodies. This approach was rooted in the concept that neutralizing antibodies could potentially protect against HIV infection by impeding the virus’s entry into target cells [1,2,3,4,5]. Nevertheless, the highly variable and mutable nature of HIV posed significant challenges in inducing broadly effective neutralizing antibody responses [6]. A subsequent wave of HIV vaccine development redirected focus towards CD8+ T cells, acknowledging their importance in controlling HIV infection. HIV-specific CD8+ T cells have the ability to eliminate target cells and respond to various HIV strains [7,8,9]. The crucial role of CD8+ T cells in preventing viral escape, along with their broad recognition capability, highlights the potential to harness them for the development of a globally effective multi-clade HIV vaccine. T cell-inducing recombinant viral vectors, such as adenovirus and poxvirus-based vectors, as well as DNA-based HIV vaccines, have been developed based on this concept [10]. During this HIV vaccine era, significant progress was made in understanding T cell responses and identifying T cell epitopes; however, it was often observed that these CD8+ T cells vaccines, although immunogenic and generally safe, could not protect against HIV acquisition [11]. T cell vector-based strategies faced additional challenges, including immune responses directed at the vector itself (rather than the HIV inserts), which diminished vaccine immunogenicity. Additionally, these strategies led to the generation of newly activated CD4+ T cells—the preferred target of HIV—accentuating the risk of infection [12].

The third wave of HIV vaccine development focused on using a dual approach to harness immune responses elicited by DNA or vector-based vaccines in combination with protein components in prime–boost strategies [13]. One such strategy quickly progressed to clinical trials and resulted in the only successful HIV efficacy vaccine trial in humans to date, although the efficacy was modest [14]. This era also emphasized the exploration of new and improved adjuvants for the protein components, as the currently licensed adjuvant, alum, was found to be less effective than non-licensed adjuvants in inducing high HIV-specific antibody titers [15,16]. However, subsequent attempts to enhance vaccine efficacy, including modifying adjuvants, have not yielded significant results in clinical settings [17,18,19].

***Recent HIV Vaccine Development***. In recent years, there has been a shift towards reevaluating the types and functions of antibodies necessary for prevention, the quality of induced T-cell responses, and the importance of boosting innate immune responses alongside adaptive-specific responses. Current research predominantly focuses on broadly neutralizing antibodies (bNAbs), which can neutralize a wide range of HIV strains by targeting conserved regions of the virus [20]. This line of research begun in 2000 but gained traction in 2009, with the development of a high-throughput HIV neutralizing assay and single-cell antibody cloning techniques [21,22]. The isolation of second-generation bNAbs has resulted in a significantly improved neutralization breadth and potency in antibodies targeting seven highly conserved major epitopes of the HIV envelope (Env) glycoprotein, including the CD4-binding site (CD4bs), the variable V1V2 and V3-glycan regions, the gp120-gp41 interface, the gp120 silent face, the gp41 conserved membrane-proximal external region (MPER), and the gp41 fusion domain [23,24,25]. The administration of bNAbs has been deemed safe and tolerable in healthy individuals [26,27,28,29,30], and the most promising bNabs are undergoing clinical trials. The discovery of second-generation broadly neutralizing antibodies has reignited interest in antibody-mediated treatment. However, these antibodies encounter challenges, including viral resistance, host viral diversity, and rapid decay when bNAbs are passively administered. Researchers are actively exploring strategies to establish bNAbs, either through administering more effective and durable bNAbs directly (passive immunization) or by designing immunogens to stimulate their production in vivo (active immunization) [31]. The specific sequence of immunogens needed to trigger broadly neutralizing antibodies has not been determined, but there is an improved understanding of the immunobiology involved in developing these potent antibodies and the challenges that impede their production (reviewed in [20]).

Additionally, RNA vaccines have shown promise in infectious diseases, including the successful development of mRNA vaccines against COVID-19 [32,33,34]. Ongoing research and development efforts are exploring the use of RNA-based vaccines for HIV [35,36]. Some of these vaccine strategies also incorporate conserved regions of the virus to enhance effectiveness against a broader range of HIV strains [35].

The evolution of HIV vaccine research reflects the dynamic nature of the field and the adaptive response to the challenges posed by the virus. The shifting focus in the HIV vaccine field underscores the need for a multidimensional approach to developing an effective vaccine. It is now recognized that harnessing multiple aspects of the immune response, including neutralizing antibodies, non-neutralizing antibody effector function, T cell immunity, and potentially other immune mechanisms such as different arms of innate immunity, may be required for comprehensive protection against HIV infection. This review provides an overview of the current landscape of HIV vaccine research, highlighting key advancements, challenges, and promising strategies on the path towards developing an effective preventive HIV vaccine.

## 2. HIV Vaccines: Clinical Trials

***Protein-based vaccines.*** The first Phase 3 HIV vaccine efficacy trials ever conducted in humans were the VAX003 (NCT00006327) and VAX004 (NCT00002441) trials; these trials utilized a vaccine formulation consisting of the bivalent glycoprotein (gp)120 envelope protein with alum adjuvant and aimed to induce anti-HIV Envelope (Env) antibodies [5,37]. VAX003 enrolled individuals who inject drugs in Thailand and utilized a bivalent envelope subtype B, along with the most prevalent genetic type of HIV-1 in Asia, initially named “subtype E” or the circulating recombinant form CRF01_AE (AIDSVAX B/E by VaxGen) [38]. In contrast, VAX004 enrolled men who have sex with men (MSM) and women at risk of HIV acquisition in North America and the Netherlands. It utilized bivalent subtype B proteins (AIDSVAX B/B). The rationale for advancing these two gp120 subunit vaccines into human trials stemmed from their capacity to elicit neutralizing antibodies against T cell line-adapted (TCLA) viruses. These viruses were obtained from a limited pool of strains and cultivated in immortalized CD4+ cell lines (e.g., IIIB/LAV, MN, SF2 in H9 or CEM) [39,40,41]. Both trials were completed successfully in 2004; however, neither trial succeeded in preventing HIV infection, reducing viral replication, or impeding disease progression. Interestingly, within the VAX003 trial, HIV-1 subtype E was responsible for 77% of infections, with 83 cases occurring among vaccine recipients and 81 among those who received the placebo. Overall, the results suggested that bivalent recombinant monomeric gp120 in alum could not confer protective efficacy against HIV. Nevertheless, the failure of these vaccines prompted further investigation in the field, focusing on inducing neutralizing antibodies by reevaluating strain selection (TCLA versus primary isolates) and the structural presentation of gp120 antigens (utilizing native trimeric spikes instead of monomeric gp120) [37,42]. Additionally, there was a reevaluation of the importance of inducing T cell responses through HIV vaccination to achieve protection from HIV acquisition and replication.

***Adenovirus Vector-based Vaccines.*** The disappointing results of the VAX003 and VAX004 trials led to a change in strategy towards developing vector-based vaccines that elicit strong HIV-specific T-cell responses [7]. The first “T-cell HIV vaccine” tested in humans was the STEP or HIV Vaccine Trial Network (HVTN) 502 trial. This Phase 2 trial was designed to evaluate a replication-defective Adenovirus serotype 5 (rAd) vectored vaccine expressing HIV Gag, Pol, and Nef antigens (Merck rAd5-gag/pol/nef) and given in three doses to MSM and sex workers in the Americas and Australia [43]. The vaccine induced a robust antiviral type I response that was measured as CD8+ T cells producing IFNg or IL2, or TNFa to vaccine-matched proteins, using intracellular staining and flow cytometry analysis [44]. Even though the STEP trial prompted the production of elevated levels of HIV-specific CD8+ T cells through vaccination, it did not show effectiveness in preventing HIV infection or decreasing viral replication in the vaccinated individuals who became infected. In fact, an unexpected outcome from the STEP trial was an elevated risk of HIV acquisition (HR). This risk was noted when considering the entire population (HR 1.4, *p* < 0.03) and was even more pronounced in a specific subset of participants who were both uncircumcised and had pre-existing immunity to the Ad5 vector used in the vaccine (HR 4.2, *p* < 0.02) [45]. In 2007, the National Institute of Allergy and Infectious Diseases (NIAID) halted the STEP trial due to safety concerns. Following the disclosure of the STEP trial findings, the Phambili trial using the Merck Ad5 vaccine platform in South Africa was halted and unblinded during the enrollment phase. Only a limited number of Phambili participants received the initially planned three vaccines. The primary analysis of the follow-up data from Phambili showed no increased risk of HIV infection [46]. However, additional insights from the extended unblinded follow-up of Phambili participants suggested a higher risk of infection among vaccinated men compared to those who were not vaccinated [47].

In vitro experiments indicated that CD4+ T cells specific to Ad5 are highly susceptible to HIV infection, and these cells are selectively diminished in individuals with HIV-1. These results prompted significant questions about the impact of pre-existing anti-vector immunity and the use of Ad5-vectored vaccines in regions where Ad5 is common. Nonetheless, in 2009, a different Ad5 vaccine platform progressed to efficacy testing. The HVTN 505 trial (NCT00865566) served as the follow-up to the STEP trials. It was a phase IIb study conducted by the HIV Vaccine Trials Network (HVTN) to assess a DNA prime/rAd5 boost vaccine. This vaccine included immunogens targeting HIV Gag, Pol, and Nef, and, in contrast to the STEP trial, also included Env. The study recruited 2504 participants at high risk of HIV infection, specifically targeting men and transgender females engaged in sex with men. To address safety concerns, MSM displaying preexisting Ad5 antibodies were excluded from the study [45]. The HVTN 505 trial was prematurely halted in April 2013 because interim analyses revealed a lack of efficacy. The results indicated that the DNA/rAd5 vaccine regimen did not decrease the rate of HIV-1 acquisition or the viral load set point in the population under study [48]. Interestingly, noteworthy inverse correlations were observed between the DNA/rAd5 vaccine-induced CD8+ T-cell immune responses to the Env protein and HIV-1 infection. Volunteers who exhibited lower levels and fewer polyfunctional Env-specific CD8+ T cells had an increased risk of HIV infection [49]. To enhance our comprehension of the influence of rAd5 vaccination on HIV acquisition, a meta-analysis incorporating data from the Step, Phambili, and HVTN 505 trials was conducted. The combined analysis showed an overall risk of 1.33 (*p* < 0.01), and when considering only the Step and Phambili studies, a hazard ratio (HR) of 1.41 (*p* = 0.005) was observed. Additionally, studies involving humans vaccinated with HVTN 505 indicated elevated levels of activated anti-rAd-specific CD4+ T cells in the gut, potentially rendering them more susceptible to HIV (McElrath JM, Mini summit on Adenovirus Platforms, 2013) [12]. Furthermore, it was shown that pre-immunization adenovirus-specific CD4+ T cells were associated with a substantially decreased magnitude of HIV-specific CD4+ T cell responses and a decreased breadth of HIV-specific CD8+ T cell responses in STEP vaccine recipients, regardless of Ad5 status [50]. It was concluded that the effectiveness, or lack thereof, of an HIV vaccine might be influenced by a delicate equilibrium between two opposing factors—protection facilitated by anti-HIV responses and the potential risk of heightened acquisition resulting from cellular activation triggered by anti-vector or any other immune-activating response.

While the continued use of rAd5-based vector vaccines for HIV prevention was deemed inappropriate [12], other adenovirus backboned vectors were under consideration or in the pipeline. Adenovirus 26 (Ad26) was considered appropriate because it is less prevalent in the human population, reducing the likelihood of pre-existing immunity, when compared to Ad5. Ad26 had showed safety in studies conducted in non-human primates [51,52]. The Ad26 “mosaic” vaccine approach was developed using viral vectors (Ad26 or modified vaccinia Ankara-MVA), protein boosts, and specially optimized immunogen sequences to create polyvalent “mosaic” antigens. Mosaic antigens were generated from natural sequences, including common B and T cell epitopes while excluding rare ones [53], to elicit broader responses against variants of HIV-1 [54,55]. The mosaic antigens, designed using in silico methods, incorporated a comprehensive array of potential epitopes derived from group M variants of HIV-1 Env, group-specific antigen (Gag), and polymerase (Pol) proteins [53].

The clinical use of these mosaic antigens draws insights from NHP studies (discussed below) and the APPROACH study (NCT02315703), which evaluated various regimens containing Ad26 or MVA vectors expressing mosaic antigens, some administered together with gp140 boosts [51]. All regimens have proved safe and well tolerated, with strong antibody responses detected. The mosaic antigens elicited binding Immunoglobulin (Ig)G responses to cross-clade transmitted/founder Envs and other variants, similar to vaccine homologous responses. Antibody-dependent cellular phagocytosis (ADCP) responses were found to be increased in the gp140-boosted groups, and serum neutralizing activity was observed against Tier-1 HIV variants [51]. Subsequent clinical trials (TRAVERSE, ASCENT, IMBOKODO, and MOSAICO) expanded on these findings, testing various formulations and regimens [56]. These trials have shown promising results in terms of safety and immunogenicity, with some formulations advancing to larger Phase 3 trials. The IMBOKODO phase 2b trial (HVTN 705, NCT03060629) tested the mosaic-based adenovirus serotype 26 vector Ad26.Mos4.HIV and the aluminum-phosphate adjuvanted Clade C gp140; however, it did not prevent HIV infection in a population of young women in sub-Saharan Africa. In a comparison of new HIV infections between participants randomly assigned to either the placebo or the investigational vaccine, statisticians observed that 63 individuals in the placebo group and 51 individuals in the experimental vaccine group contracted HIV infection. The vaccine efficacy was 25% (with a 95% confidence interval of vaccine efficacy −10.5% to 49.3%). The phase 3 trial MOSAICO (HVTN 706) was tested among men who have sex with men (MSM) and transgender people, involving 3900 volunteers aged 18 to 60 years in Europe, North America, and South America. The vaccine regimen was Ad26.Mos4.HIV administered during four vaccinations in one year. Clade C/Mosaic gp140, adjuvanted with aluminum phosphate, was also administered at visits three and four. This trial also proved ineffective and was discontinued in January of 2023. Currently, other Ad-based vectors are under investigation, including Ad35 (NCT01264445, phase I) [57] and Ad4 (NCT01989533, phase I) [58]. Moreover, the mosaic concept has been extended to a DNA prime MVA boost HIV vaccine strategy, marking the first test of a trivalent mosaic approach in humans [59]. In a trial involving 105 healthy individuals (HVTN 106), Cohen and colleagues investigated variations in the breadth and levels of cellular responses to native, consensus, and mosaic antigens. They discovered that mosaic antigens exhibited superior breadth in immune responses. However, it is worth noting that most of this diversity was attributed not to CD8+ T cells but to Env-specific CD4+ T cells instead [59]. The authors concluded that using mosaic antigens for priming significantly increased the number of epitopes recognized by Env-specific T cells, leading to the enhanced, albeit still limited, cross-recognition of heterologous variants. Due to the substantial differences between IMBOKIDO and HTV 106, comparing the results from these trials is not feasible. However, considering the mosaic Ad26-based trial’s lack of efficacy and the limited cross-reactivity observed in HTVN 106, it is evident that there is a need to enhance the capacity of mosaic strategies to induce protective cellular responses against heterologous circulating strains.

***ALVAC Vector-based Vaccines***. The RV144 Phase III HIV-1 vaccine trial was conducted in Thailand from 2003 to 2009 [14,60]. Enrolling over 16,000 participants from the general population, the trial was a collaborative effort between the Thai Ministry of Public Health, the US Army Surgeon General, the US Military HIV Research Program (MHRP), and various Thai and US government agencies, private companies, and nonprofit organizations [61]. The vaccine regimen employed in the RV144 trial involved a prime–boost strategy, combining two vaccines: ALVAC-HIV from Sanofi Pasteur and AIDSVAX B/E from VaxGene, the same protein boost used in VAX003 adjuvanted in alum. These vaccines were designed based on the HIV-1 B and E clades prevalent in Thailand [62]. The vaccine’s objective was to stimulate T-cell responses and antibodies targeting clades B and E. The ALVAC-HIV, a virus recombinant Canarypox vCP1521-based T cell vaccine, was designed to induce cell-mediated immunity [37,63]. Simultaneously, the AIDSVAX B/E bivalent gp120 was anticipated to generate substantial levels of neutralizing antibodies in Thai subjects [37]. The two vaccines were given with a prime–boost approach [13]. The selection criteria for their dose were determined by evaluating the vaccines’ component reactogenicities and manufacturing feasibility, as demonstrated in several Phase I/II trials with ALVAC-HIV [64,65,66,67,68,69,70] and the Phase III VaxGen trial [5,62]. The primary objective of the trial was to evaluate the efficacy of this combination in preventing HIV infection and reducing viral RNA levels in infected individuals. There were initial doubts and debates surrounding the immune response generated by this vaccine combination [71,72], given the relatively low immunogenicity of ALVAC compared to other vector-based strategies. Nonetheless, the RV144 trial demonstrated its safety and showed 60% vaccine efficacy (VE) at 12 months post immunization, which decreased to 31% at 3.5 years [14]. While the level of efficacy was modest, the results provided encouraging evidence for the feasibility of an HIV vaccine and indicated that further research is necessary to develop a vaccine capable of effectively safeguarding the general population against HIV acquisition. Notably, this vaccine provided the opportunity to study correlates of protection in the context of an effective vaccine for the first time. The examination of correlates encompassed antibody responses, T cell responses, and breakthrough viruses. It was reported that neutralizing antibodies against circulating Tier 2 HIV-1 strains from Thailand were undetectable in the RV144 trial, suggesting that the modest efficacy was largely attributed to non-neutralizing antibody effector functions [41,73,74,75,76]. The high avidity of IgG antibodies for the envelope, as well as the antibody-dependent cellular cytotoxicity (ADCC) and phagocytosis (ADCP) activities, were inversely correlated with the risk of infection [74,77,78,79,80,81]. These data suggest that the non-neutralizing effector function of antibodies may play a role in preventing HIV acquisition. Specifically, IgG antibody binding to the V1V2 region of the envelope demonstrated an inverse correlation with the infection rate, indicating a potential protective effect. In contrast, binding IgA antibodies to the envelope showed a direct correlation with the rate of infection, suggesting a potential detrimental impact on vaccine efficacy [73,76]. A genomic sieve analysis, focusing on the V1V2 region of Env and comparing breakthrough HIV-1 sequences between the infected vaccine and placebo groups, pinpointed two sites in the V2 loop (amino acid positions 169 and 181) associated with efficacy [82]. Subsequent sieve analysis also identified potential immune pressure in the V3 loop of the HIV-1 envelope [76]. These findings underscore the significance of evaluating both the quality and specificity of antibody responses in HIV prevention [83].

The RV144 vaccine regimen induced modest ex vivo T cell responses with respect to frequency and magnitude, as measured by IFNg ELISPOT and intracellular staining (ICS) by flow cytometry analysis, compared to other HIV vaccine regimens with either poxvirus [84] or adenovirus [54,55,85] vectors. In RV144, CD8+ cytolytic T cells were detectable using the chromium release assay; however, direct ex vivo HIV-specific CD8+ T cell responses from RV144 subjects’ peripheral blood mononuclear cells (PBMC) were scarcely measurable (less than 10%) in the IFNg/IL-2 combination ICS assay [14,68]. These responses were comparable to the frequency observed in placebo recipients. In contrast, substantial CD4+ T cell responses, evaluated through both [3H] incorporation and ICS assays, were reported in the vaccine group [14]. The CD4+ T cell response targeted the HIV-1 Env, specifically focusing on the V2 variable loop region og the gp120. This region encompasses the presumed alpha4beta7 binding site, an integrin crucial for establishing virological synapses on CD4+ T cells in the gut [86,87,88].

Post hoc analyses unveiled significant correlations between the binding antibody responses and CD4+ T cell responses affecting the rate of HIV acquisition [86,89,90]. Additionally, the presence of Env-specific CD4+ T cells was found to be inversely correlated with the risk of infection, further emphasizing the importance of cellular immune responses in vaccine-induced protection [89]. The polyfunctional response in Env-specific CD4+ T cells expressing CD154 (or CD40 ligand) and secreting cytokines such as IL-2, IL-4, IFNg, and TNFa demonstrated the most robust correlation, resulting in a lower infection rate compared to individuals who did not generate such a multifaceted immune response [89,91]. Finally, a transcriptomic analysis of RV144 trial samples identified a role for the interferon regulatory factor 7 or IRF7 as a mediator of protection and showed that the activation of mTORC1 was a correlate of the risk of HIV-1 acquisition [90,92].

Two early-phase trials, RV305 and RV306, aimed to enhance the durability of the immune responses observed in the RV144 trial. RV305 enrolled individuals who had previously received the RV144 vaccine, investigating the effects of additional boosting with ALVAC-HIV and AIDSVAX B/E. The results showed that boosting RV144 would evoke memory responses, inducing higher levels of IgG responses against gp120 and gp70-V1V2 compared to peak immunogenicity in RV144. Moreover, subsequent booster vaccinations resulted in the production of antibodies exhibiting characteristics similar to broadly neutralizing antibodies, including increased somatic hypermutation and a longer immunoglobulin heavy-chain complementarity-determining region 3 (HCDR3) length. However, it was also observed that repeated boosting skewed the responses towards the IgG4 subclass, which is associated with reduced non-neutralizing function, and did not improve the durability of antibody responses [93,94,95,96].

HVTN 097 was conducted in South Africa and utilized the same formulation, prime–boost concept and immunization schedule as the RV144 trial [97]. The trial aimed to prospectively assess the correlates of infection risk and explore potential cross-clade immune responses induced in South African individuals when using the RV144 regimen, which includes clade B and E inserts, in a region dominated by clade C. Responses were compared to those elicited in RV144 and included the overall response rates of plasma IgG and Env-specific CD4+ T cells expressing IFNg and/or IL-2. The results of HVTN 097 showed that the response rates were similar to those observed in RV144 [97]. This trial was a precursor to an adapted regimen, involving the subtype C ALVAC-HIV-1 and bivalent subtype C gp120/MF59 HIV-1 vaccine regimen HVTN 100 [98].

HVTN 100 was also conducted in South Africa and employed a pox-protein vaccine regimen specifically designed for the local subtype C epidemic. The vaccine regimen consisted of the ALVAC-HIV vCP2438, which expressed HIV subtype C gp120, subtype B gp41, gag, and protease, followed by a boost with a bivalent subtype C (TV1/1086) gp120. Additionally, an alternative adjuvant, the MF59 oil-in-water emulsion, was used instead of the aluminum hydroxide adjuvant used in RV144. The primary objectives of HVTN 100 were to evaluate the safety and tolerability of the vaccine regimen and assess the immune responses elicited by the vaccine [99]. The study found that all vaccine recipients developed gp120-binding antibodies, and that these antibody levels were significantly increased compared to RV144. Furthermore, the vaccine regimen induced higher CD4+ T cell responses to the corresponding envelope protein. Although the IgG antibody responses directed at 1086_V1V2 were lower in HVTN 100 compared to the RV144 regimen in HVTN 097 [83,100], the results showed that the vaccine met the criteria for advancing to the next phase, which was the HVTN 702 trial. The HVTN 702 Uhambo (“journey”) 2b/3 efficacy trial began in 2016 and enrolled individuals at risk of HIV in South Africa. However, results from the interim analysis revealed no significant evidence of changes in the infection rates associated with the vaccine regimen. Consequently, the trial was halted in February 2020 by the NIH US Data and Safety Monitoring Board for futility [19,98,101]. Considerations of the many distinctions between the South African and RV144 trial are important to prevent any mistaken inference that the results of the former trial undermine those of the latter [102]; these differences include (1) the vaccine vector, the protein (gp120 vs. gp160 and Clade BE vs. C) and the adjuvant components (alum vs. MF59), (2) the composition and vulnerability of the population studied (75% women at high risk in Africa compared to population-based risk in Thailand), (3) a higher likelihood of a mismatch between the vaccine and the circulating strains and (4) differences in pre-existing immunity in South Africa compared to the Thai setting. Finally, the HVTN 111 trial employed a DNA-prime strategy (subtype C DNA-HIV-PT123) and the same gp120 boost used in HVTN 100 (bivalent subtype C gp120/MF59). This trial resulted in increased immune responses when compared to HVTN 100, including increased levels of CD4+ T cells and binding and neutralizing antibodies [103]. The enhanced immunogenicity of HTVN 111 is deemed too modest to justify further advancement. However, it is acknowledged as a preliminary step in refining HTVN702 and laying the groundwork for optimizing an ALVAC-based vaccine with efficacy levels surpassing those observed in RV144.

***Broadly neutralizing antibodies*.** In the early days of the HIV epidemic, studies found a link between high neutralizing antibody levels and delayed disease progression in people with HIV (PWH) [104,105]. This discovery led to experimental transfers of hyperimmune plasma to individuals with active virus replication. Achieving control of HIV infection through vaccination by inducing neutralizing antibodies has since been a primary goal of the HIV vaccine field [106,107]. However, while the first generation of gp120 protein-based vaccines were safe and generated neutralizing antibodies in clinical trials, they did not effectively prevent HIV-1 infection, as in the case of VAX003 and VAX004. By utilizing multiclade Env panels and serum samples, it was later revealed that HIV-1 isolates display varying degrees of neutralization sensitivity, categorized into four tiers [40,108]. Tier 1A represents the most sensitive phenotype, followed by Tier 1B, Tier 2 (most circulating strains), and Tier 3 (the least sensitive) (reviewed in [109]). While some Env immunogens elicit antibodies that neutralize Tier 1A and 1B Envs, they often fail to neutralize Tier 2 and 3 Envs, challenging the utility of Tier 1A neutralization as a benchmark for HIV-1 vaccines [109].

Advances in antibody isolation and cloning techniques, including improved antigen design and B-cell receptor amplification, have enabled the identification of highly potent antibodies capable of neutralizing a wide range of HIV strains. Over the past decade, more than 60 clinical trials have explored the pharmacokinetics and immunological effects of these broadly neutralizing antibodies in humans. Currently, researchers are actively investigating strategies to elicit bNAbs through vaccination, either by administering bNAbs directly or designing immunogens to stimulate their production [110]. The Assessing Antibody-Mediated Protection AMP trial investigated the potential of the long-term administration of the passively infused bNAbs VRC01, targeting the CD4-binding site (CD4bs), to prevent HIV-1 acquisition in humans (HVTN 704/HPTN 085 and HVTN 703/HPTN 081) [27]. It involved 4600 at-risk participants from diverse geographical regions, including Sub-Saharan Africa, the Americas, and Europe. The results revealed that VRC01 could prevent HIV-1 infection, with a high prevention efficacy of 75% observed against viruses sensitive to VRC01 (IC80 < 1 μg/mL). However, there was no efficacy against most circulating strains (with IC80 values > 1 μg/mL), resulting in no significant overall protection. Interestingly, the outcome was reminiscent of what was observed with first-generation antiretroviral therapy, where innate resistance and the emergence of resistant isolates over time compromised the effectiveness of single therapeutic agents for prevention, thus suggesting the necessity of evaluating the efficacy of a more comprehensive and potent combination of antibodies [27].

Efforts to achieve bNAbs through active immunization are also ongoing. The difficulties experienced in eliciting protective bNAbs can be attributed to various HIV-1 immune evasion strategies, including antigenic diversification during replication and the dense glycan shield on Env that hides critical antigenic epitopes from the immune system [71]. The structural dynamics of the Env trimer, with its distinct conformations, trigger different antibody responses. The closed prefusion conformation is recognized by potent bNAbs, while antibodies targeting regions exposed in the open conformation induced by CD4 binding are weak or non-neutralizing and are ineffective at preventing infection. Using a structure-based vaccine design, stabilized viral immunogens that remain in the closed prefusion conformation and can generate protective antibodies have been developed. SOSIPs are uniform, soluble, stable and trimeric forms of the HIV-1 envelope spike that closely resemble the native viral spike in terms of antigenicity and structure. They achieve stability through a disulfide bond called “SOS” between gp120 and gp41 and a specific point mutation named “IP” at residue 559, which helps maintain their trimeric structure [111,112]. This approach has been successful in creating vaccines against other viruses [113]. For HIV-1, a soluble protein trimer immunogen was designed based on the clade A HIV strain BG505 (BG505 SOSIP.664) [111,114,115]. Prior studies showed that this construct, although containing stabilizing mutations, could still be recognized by non-neutralizing, CD4-induced antibodies. An additional disulfide mutation (DS) was introduced within gp120 to prevent any CD4-induced conformational change. This modified prefusion-closed conformation immunogen, Trimer 4571 (BG505 DS-SOSIP.664), exhibited the desired antigenic profile and was resistant to CD4-induced conformational changes [116]. A phase 1 small sample size clinical trial concluded with encouraging results [116].

## 3. Preclinical Evaluation of HIV Vaccines in NHPs

Having elucidated the landscape of HIV vaccine clinical trials in humans in the first part of this review, it is pivotal to delve into the foundational studies conducted in non-human primate models. The vaccines discussed earlier underwent rigorous testing in these models, providing essential insights into their safety, immunogenicity, and potential efficacy. We will now transition to the preceding studies that paved the way for these vaccines to be tested in human trials, as well as explore preclinical studies that, while promising, await evaluation in human subjects.

There is no doubt that non-human primate (NHP) models provide valuable insights into potential strategies for preventing HIV, playing a crucial role in informing and guiding the development of new vaccine candidates before they advance to human clinical trials. The history of macaque models in HIV vaccine research can be traced back to the early 1980s when scientists began searching for an animal model that could mimic the immunopathogenesis of HIV infection. Apes such gibbons and chimpanzee are susceptible to HIV infection but were not deemed reasonable hosts [117], while Asian macaques, including *Macaca mulatta* (rhesus macaques, RM) and *M. fascicularis* (cynomolgus monkeys), were reasonable models but the cells from these species appeared to be resistant to HIV-1 [118]. In 1984, a breakthrough was the discovery of a related lentivirus, simian immunodeficiency virus (SIV), in captive macaques. SIV naturally infects various African non-human primates, such as sooty mangabeys and African green monkeys, without inducing disease; however, it leads to immunodeficiency in Asian macaques that are experimentally infected [119,120,121,122]. The similarity between SIV and HIV, as well as the ability of SIV to cause an AIDS-like disease in macaques, led to the establishment of SIV infection models in macaques to study the pathogenesis and immune responses associated with HIV infection. Macaques, specifically Indian-origin rhesus macaques (*Macaca mulatta*) and pig-tailed macaques (*Macaca nemestrina*), have been widely employed due to their close genetic and immunological similarities to humans, making them valuable surrogate models for studying HIV infection and vaccine responses. Macaque models of SIV infection have been extensively utilized to assess the safety and immunogenicity of potential HIV vaccine candidates before moving to human clinical trials [123].

Some of the earliest SIV vaccine studies in NHPs involved using inactivated SIV in adjuvant, demonstrating a partial protective effect against high-dose pathogenic SIV challenge [124]. Notably, a formalin-inactivated SIV vaccine showed impressive results, protecting eight out of nine vaccinated rhesus macaques against pathogenic challenge [125]. Encouraging outcomes from several studies employing different approaches to whole-virus inactivation, as well as fixed virus-infected cell preparations, fostered considerable optimism about the future of AIDS vaccines [126]. However, subsequent research revealed that the protection was associated with antibodies to the host cellular proteins in the virus generated in human cell lines, rather than with anti-SIV immune responses. Rhesus macaques immunized with fixed SIV-infected cells of human origin exhibited evidence of a xenogeneic immune response against the human cell line in which the virus was cultivated, including antibodies to HLA class I molecules [127,128]. Immunization with purified HLA class I and class II antigens demonstrated protection against challenge with SIV stocks grown on human cells but not on rhesus cells, indicating that protection was mediated by the generation of anti-MHC antibodies binding to the human HLA molecules present on the surface of SIV grown in human cells [127,129]. These early setbacks underscored some of the challenges facing AIDS vaccine development. Among the various approaches tested in the NHP model, live attenuated SIV vaccines have consistently provided the most reliable protection against pathogenic SIV isolates to date. Despite concerns regarding potential virulence, NHP models of live attenuated SIV infection offer opportunities to investigate the immune correlates of protection that can inform the development of a viable AIDS vaccine [129,130].

***Preferred NHP models for HIV vaccines***. There are currently multiple NHP (non-human primate) models used in HIV vaccine studies, varying in species, challenge route, virus dose, and strain. Rhesus Macaques (RMs), especially of Indian-origin (in contrast to Chinese-origin), are prominently utilized in these studies in the US. It is notable that Indian-origin RMs often exhibit lower viral loads compared to Chinese-origin RMs, both in the acute and chronic stages of infection [131,132]. While this difference may be seen as a potential limitation, researchers have successfully addressed this challenge by creating alternative viruses that are specifically adapted for RMs. Pig-tailed macaques have also been used; however, they have higher baseline immune activation levels than RMs even without infection and differ in their HIV-1 restriction factors (TRIM5alpha) [133,134]. Nonetheless, studies in Pig-tailed macaques offer a good model for intravaginal challenge because females possess menstrual cycles like humans, making them valuable for studying factors affecting susceptibility to vaginal infection and evaluating interventions for preventing vaginal virus transmission.

***Virus challenge***. An important focus in vaccine studies involving macaques is the selection of the right virus for challenging NHPs. Choosing a virus with high virulence and strong replication can lead to excessive pathology, overwhelming the host’s immune responses after vaccination and resulting in an underestimate of vaccine efficacy. Conversely, using a virus with low virulence and weak replication might be easily controlled by the vaccine-induced immune response, leading to an overestimate of efficacy [135,136,137]. Additionally, some viral strains are highly sensitive to neutralizing antibodies, making them unsuitable for evaluating mucosal transmissions. Fortunately, many challenge viruses have been developed, offering a range of options for preclinical vaccine studies using NHP models, and these viruses have been well reviewed and summarized elsewhere [138]. SIVmac251 and SIVmac239 are among the most widely used viruses in early NHP vaccine studies. SIVmac251 is a swarm that was first isolated from rhesus macaque “251” at the New England Primate research center [139]. Over time, various SIVmac251 stocks have been generated through in vitro passage or by isolating new viral populations from infected animals. These stocks consist of heterogeneous swarms of viruses that can transmit multiple variants across mucosal tissues [140,141,142]. This diversity is important in NHP vaccine studies, as it reflects the complexity of viral populations encountered in natural infections and provides a more realistic model for evaluating vaccine efficacy. When infected with pathogenic SIV strains, RMs exhibit consistent disease progression and high viral loads during the acute phase of infection, allowing the impact of vaccination on viral replication, disease progression, and immune system dynamics to be studied. The genetic variations observed in the reverse transcriptase and protease of HIV-1 and SIVmac pose challenges in assessing the effectiveness of antiretroviral drugs that specifically target these proteins within the SIVmac-RM model. Additionally, evaluating antibody-based vaccines against HIV-1 using this model becomes impractical due to differences in the HIV-1 and SIV envelope and the lack of cross-reactivity in neutralizing antibodies. To address these limitations, researchers have developed chimeric viruses known as SHIVs, which incorporate specific HIV genes (HIV-1 *env* along with accessory genes *tat*, *vpu* and *rev*) into the SIV backbone [19,143,144,145]. Recent initiatives have shifted towards employing transmitted/founder (T/F) HIV-1 Env clones, which are highly pertinent Envs for transmission research and vaccine evaluations [140,144]. These T/F SHIVs can facilitate mucosal transmission and trigger strong viral replication in rhesus macaques without the need for consecutive modifications. Ongoing endeavors are directed at creating SHIVs with greater viral diversity and neutralization characteristics, as well as consistently sustaining elevated chronic viral loads and progressive infection patterns, to more accurately replicate HIV pathogenesis in specific prevention studies.

***Challenge route.*** NHP models have been employed to study various routes of virus transmission related to HIV-1, including intravenous, intrarectal, intravaginal, penile, oral, and intrauterine transmission, mimicking various modes of HIV transmission. However, these models have limitations in accurately mirroring human transmission scenarios. The intravenous route for HIV challenge is considered the most reliable in NHPs. However, it has reduced clinical relevance for prophylactic vaccines that aim to prevent mucosal HIV transmission. This route does not replicate relevant aspects of HIV mucosal transmission in humans, which is how the virus commonly spreads. In preclinical studies involving macaques, a significant focus has been on mucosal challenges since HIV infection is often acquired through heterosexual transmission via mucosal exposures. Historically and currently, intrarectal challenges are commonly used because they provide a relatively easy means of infection and allow for the use of both male and female NHP. In contrast, intravaginal challenges were initially less frequently employed, mainly because of the limited availability of female macaques (sustaining the breeding colonies in Non-Human Primate Centers across the US). Moreover, the menstrual cycle, vaginal mucosal structure, and microbial composition all play roles in influencing susceptibility to SIV or SHIV infection. Despite the challenges and limitations associated with intravaginal challenges, preclinical trials using female macaques have increased in number in recent years. This method is considered the best way to simulate the male-to-female HIV transmission route, which accounts for most HIV transmissions in humans. Penile challenges have also been developed in macaques; however, similar to intravaginal transmission, they require higher virus doses and exhibit more variability between animals compared to intrarectal transmission [140,142,146,147,148].

***Virus dose*.** In the past, researchers used higher virus doses, approximating the minimum amount required to infect most unvaccinated control animals with a single challenge. This approach allows for assessing vaccine effectiveness through sequencing techniques, along with evaluating infection rates, the number of transmitted variants per animal, and performing a sieving analysis. However, the HIV transmission rate per coital act is estimated to be very low in humans [149,150]. It was also discovered that in humans, infection results from a limited range of viral variants responsible for systemic infection following sexual transmission of HIV-1, typically involving 1 to 5 T/F [140,142,151]. To better replicate human transmission, preclinical studies in NHPs have shifted their focus towards utilizing repeated low-dose challenge paradigms [152,153,154,155]. This involves determining a challenge dose that infects only a portion of unvaccinated control animals per exposure and subjecting animals to repeated exposures until signs of infection emerge or a predetermined number of challenges is reached. While the inoculum size used in these studies may still exceed typical human exposures, it helps simulate human mucosal transmission by emulating a restricted set of initial viral variants. It is important to note that variations in challenge modalities, including the dose, can significantly impact vaccine efficacy [152,156]. Nonetheless, studies using different macaque species are vital for HIV research, shedding light on pathogenesis, immune responses, and vaccine development. Each species has unique strengths and weaknesses, enabling researchers to choose the best model for specific goals. These varied models provide valuable insights, driving progress in HIV research and efforts to develop effective preventive and therapeutic measures.

## 4. Correlates of Protection in NHP HIV Vaccine Studies

Macaques play a crucial role in HIV vaccine development, serving as essential models in preclinical studies using chimeric viruses in rhesus macaques as benchmarks. The remarkable similarity between human and macaque immune systems makes macaque models invaluable for immunogenicity studies in vaccine development. These studies have demonstrated high reliability in assessing various aspects of vaccine performance in this model, offering insights into the vaccine’s safety profile and its ability to induce immune responses, regardless of the specific vaccine type under examination. While macaques are irreplaceable for evaluating vaccine immunogenicity, it is important to note that there have been disparities in vaccine efficacy between preclinical and clinical trials [133]. Recognizing these disparities is crucial for refining the model’s predictability, even though the ultimate test for a vaccine lies in human trials.

***NHP prediction of Vaccine Efficacy***. Early vaccines studies involving NHPs offered a glimmer of hope for a preventive vaccine able to control viral replication and acquisition against high-dose mucosal challenges. Studies conducted in the late 1980s and early 1990s initially generated excitement, as recombinant live and DNA vaccines demonstrated measurable CD8+ T lymphocyte cytotoxic responses (CTL) in both rhesus macaques and humans [157,158]. Additionally, several studies have demonstrated that passively administered antibodies can protect non-human primates (NHP) from simian–human immunodeficiency virus (SHIV) infection [159,160,161,162]. In monkeys, recombinant protein- and peptide-based vaccines successfully generated measurable levels of neutralizing antibodies. However, Phase III clinical trials presented an interesting contrast: recombinant HIV envelope (Env)-expressing vaccines were not as effective in eliciting broad-spectrum protective antibodies, even against closely related viruses [26]. This difference highlighted a notable distinction between the results observed in macaques and humans, prompting a re-evaluation of expectations regarding the rapid development of an HIV vaccine through neutralizing antibodies [37].

***Broadly Nabs/SOSIPS in NHP.*** In preclinical studies, immunizayion with SOSIPS was found to be safe and induced neutralizing antibodies in rhesus macaques when administered with an adjuvant. When used in combination vaccine regimens with an HIV-1 fusion peptide-coupled carrier, Trimer 4571 resulted in cross-clade neutralizing antibodies in mice, guinea pigs, and rhesus macaques [163,164,165]. Native-like SOSIP trimers have been successful in eliciting antibodies capable of neutralizing autologous tier 2 strains in animal models and rhesus macaques [166,167,168,169]. A recent study suggested that high levels of serum neutralizing antibody (nAb) titers elicited by the BG505 SOSIP trimer were linked to protection against repeated, low-dose rectal challenges with SHIV.BG505. However, the study had limitations, and the protective efficacy was not conclusively demonstrated. In response, a preclinical efficacy trial was conducted using rhesus macaques. The macaques were immunized with BG505 SOSIP in 3M-052 adjuvant alone or in combination with three different heterologous viral vectors expressing SIVmac239 Gag. These viral vectors did not express Env and were included to investigate whether anti-Gag T cell responses played a role in protection. A control group received only the 3M-052 adjuvant. The results showed significant and robust protection against repeated low-dose intravaginal challenges with SHIV.BG505 in both vaccination groups compared to the control group. Specifically, a serum neutralizing antibody titer greater than 1:319, measured two weeks after the final immunization, was found to be a reliable predictor of protection. This finding prompted further examination of the specificities and characteristics of these neutralizing antibodies associated with preventing SHIV acquisition [170]. Another study investigated the targets of nAb in rhesus macaques immunized with BG505 SOSIP and found that these nAbs predominantly targeted the 465-glycan hole cluster. Longitudinal analysis revealed that N611 antibodies emerged before nAb in some macaques, and when nAb remained focused on the 465 glycan hole, it led to an increase in nAb titers. Monoclonal antibodies from a protected macaque showed potent neutralization of BG505 Env and BG505.HIV, providing valuable insights into the immunogenicity of the C3/465 glycan hole cluster in BG505 SOSIP [171].

***Ad5–based vaccines in NHP***. Consequently, many researchers shifted their focus toward developing immunization approaches centered on harnessing antiviral T cell responses [44,172,173]. Initial findings in non-human primates indicated that these responses could potentially partially restrict infections with various SIV strains. Researchers also focused on developing robust vector systems to induce HIV-specific CTL (cytotoxic T lymphocyte) responses, with the goal of halting the spread of the virus at the mucosal level. While these second-generation vaccines, including adenovirus-based vectors, demonstrated promise in preclinical studies in macaques [44,172,173], concerns emerged regarding their efficacy against pathogenic SIV challenges in outbred genetic haplotypes [172]. Despite debates about discrepancies in studies employing different SIV and SHIV challenge models and debates about the merits and drawbacks of such models, Ad5 studies advanced into clinical trials based on the assumption that protection in macaques would translate to humans, and that non-protection would likewise correlate. However, as previously discussed, the STEP trial lacked efficacy and saw unexpected increased HIV transmission rates in certain Ad5-seropositive vaccine recipients, in stark contrast to earlier macaque studies [174].

***Ad26–based and mosaic vaccines in NHP***. Ad26-based vectors were developed and evaluated in macaques based on the assumption that humans had not been previously exposed to Ad26, unlike Ad5. Preclinical testing in macaques involved high-dose challenges with SIVs and SHIVs, and it demonstrated protection against high viral replication and mucosal acquisition when administered alone or in combination with DNA or the gp140 protein in prime–boost regimens [175,176]. However, clinical trials in humans using similar approaches did not show significant protection.

Polyvalent mosaic antigens expressed by the recombinant, replication-incompetent adenovirus serotype 26 vectors were also tested in rhesus monkeys and informed the clinical trials that followed [55,177]. Indeed, they showed that adding mosaic antigens could markedly augment both the breadth and depth without compromising the magnitude of antigen-specific T cell responses when compared with consensus or natural sequence HIV-1 antigens [55,177]. Contrary to what was later observed in humans, Ad26 mosaic vaccines protected macaques from acquisition against heterologous SHIV challenges [154,167]. Parallel to the APPROACH study, an NHP study was conducted, observing similar immunogenicity [51]. The Ad26.Mos.HIV/gp140 vaccine (adjuvanted in aluminum phosphate) showed significant protection against intrarectal challenges with SHIV-SF162P3, with an impressive 94% reduction in the per acquisition risk and 66% complete protection. Despite not recapitulating transmitter/founder HIV-1 isolates, the CCR5-tropism, relative neutralization resistance and pathogenic properties of SHIV-SF162P3 justify its extensive use as a validated challenge virus in vaccine studies [178].

***ALVAC–based vaccines in NHP*.** Together with other poxviruses, the canarypox ALVAC vector has been extensively studied in macaques against SIV, SHIV, and HIV isolates [179,180,181,182,183,184]. The results showed variable levels of cellular immune responses and the prevention of infection against HIV-2 and other attenuated SIV viruses. As with the Ad-based vectored vaccines, poxviruses also significantly reduced the peak viral loads during acute infection [181,183,184,185]. Among the pox vector-based vaccines, only ALVAC-based HIV-1 vaccines have been tested in phase 3 clinical trials and have been shown to be safe, immunogenic and partially effective in humans. The reduced protection against HIV acquisition provided by the ALVAC–SIV + gp120 alum regimen was both limited and temporary, indicating a need for enhancement. In macaques, vaccination with a comparable SIV-based vaccine regimen also notably reduced the risk of acquiring SIVmac251 (with 44% efficacy), and this effect was linked to the quantity of mucosal antibodies targeting V2 [18]. The substitution of the alum with the MF59 adjuvant resulted in a loss of vaccine efficacy, similar to what observed in the RV144 follow-up trials in Africa [19]. Taken together, these results imply that certain NHP models may offer more accurate predictions of vaccine efficacy than others, though understanding the reasons for this discrepancy is challenging. One potential factor could be the use of different virus stocks or variations in the study design. Considerable differences exist in the dosages of uncloned SIV or SHIV used across studies, including viral stocks from various laboratories with distinct passage histories and in vitro production methods. Another important consideration is that, given that around 80% of HIV transmissions result from a single highly virulent virion, employing a virus stock with either high or low variance may compromise the accuracy of modeling natural HIV infection [186]. However, a side by side comparison of the viral stock and same vaccines has never been performed because it is too costly. Another possible consideration regarding the discrepancy between macaques and humans is how vaccine efficacy (VE) is calculated. There has been increased attention given to the specific methodologies and considerations involved in assessing VE, highlighting the need for more comprehensive research and discussion in the field to establish standardized protocols and guidelines for evaluating vaccine efficacy in NHP models. Survival analyses, particularly employing Cox’s proportional hazard models and likelihood ratio tests, are the preferred methods for assessing vaccine efficacy in macaque models and comparing the risk of SIV/SHIV infection between vaccinated and unvaccinated groups over time. These analyses help define vaccine efficacy as the relative reduction in the per-contact transmission probability when comparing vaccinated and unvaccinated macaques [155,187].

What is clear is that the evaluation of vaccine efficacy is significantly influenced by the design of the challenge experiments. The design of the challenge experiments significantly impacts the efficacy assessment, including the endpoints, sample size, unvaccinated macaque infection rates, susceptible macaque proportions, and statistical methods. Precise sample size calculation to achieve at least 80% statistical power is essential. These NHP models can be further refined for evaluating HIV-1 vaccine candidates and guiding clinical trials. Overall, the macaque model’s ability to replicate human-like immune responses and safety profiles in immunogenicity studies plays a pivotal role in the development and evaluation of vaccines, helping to identify promising candidates for further clinical testing in human trials. Lastly, when possible, it is important to bridge preclinical and clinical data.

***Other HIV Vaccine Strategies Tested in NHP*.** To date, numerous studies have been conducted involving other attenuated recombinant poxvirus vectors expressing HIV/SIV antigens, with modified Vaccinia Ankara (MVA) and New York Vaccinia (NYVAC) in particular [188,189,190,191,192]. These strategies are reviewed elsewhere and are currently being evaluated in various stages of clinical trials in order to establish their efficacy [193].

The utilization of the human cytomegalovirus (CMV) vector represents a promising and innovative approach in the development of HIV vaccines. The immunization of non-human primates (NHPs) with CMV/SIV vectors has resulted in persistent and high-frequency SIV-specific memory T-cell responses at potential SIV replication sites. This resulted in the sustained control of SIV infection in 50% of the NHPs, and this protective effect was associated with the elicitation of unconventional MHC-E-restricted CD8+ T-cell responses [194]. Currently, an initial clinical trial is underway to evaluate the safety and immunogenicity of a CMV vector-based vaccine named VIR-1111, with recruitment targeting healthy individuals who are CMV seropositive [195,196]. HIV-RNA-based vaccines have also been tested and, following the success of vaccination for COVID-19, these affords have been expanding, with different messenger (m)RNA vaccines being tested [35]. Currently, mRNA HIV vaccine candidates developed by the Scripps Research Institute and Moderna are being tested in Phase 1 clinical trials [197]. A mRNA vaccine co-expressing multi-clade HIV-1 Envelope and SIV Gag proteins to generate virus-like particles elicited broadly neutralizing antibodies in macaques and resulted in a 79% per-exposure risk reduction after mucosal SHIV AD8 challenge [198].

## 5. Vaccine Induced Correlates of HIV in Humans and NHP

Correlates of protection serve as critical immune biomarkers, helping to evaluate vaccination response and predicting the anticipated level of vaccine efficacy for a specific clinical outcome [199]. Whether mechanistic or non-mechanistic, a correlate of protection is valuable as a surrogate endpoint in this context. The identification of immunological markers associated with the risk of transmission in both preclinical and clinical trials for HIV-1 vaccines has significantly propelled the field of HIV-1 vaccine development, guiding the exploration of new vaccine candidates [200]. Studies on immune correlates have spawned innovative hypotheses about the immunological processes that may contribute to averting HIV-1 acquisition. Recent research into HIV-1 immune correlates has uncovered that diverse types of immune responses may together form an immune correlate, underscoring the significance of polyfunctional immune control in averting HIV-1 transmission. A consideration of the study population and species is crucial in understanding vaccine correlates. Although various non-human primate (NHP) challenge studies, employing diverse vaccine approaches, have shown partial protection against SIV or SHIV acquisition through CD8 T cell responses and neutralizing antibodies, the only partially effective trial against HIV did not yield similar results. The discussion on the role of adaptive immune responses in protecting against HIV has been extensive; hence, we discuss the contribution of innate immune responses and preexisting immunity.

***Innate immunity.*** It is well recognized that innate responses contribute to protection from HIV-1 infection. It is also well established that DNA and viral vectors used as backbones of HIV vaccines induce innate immune responses that in turn influence the efficacy and duration of specific T cell responses. For example, replicating Adenovirus vectors, including Ad5 and Ad26, signal through MyD88 dependent and independent pathways [201], and poxvirus vectors have been shown to activate both TLR-dependent and TLR-independent innate pathways. Studies in mice have demonstrated that the vaccinia virus triggers innate immunity through both a TLR-2-dependent pathway and a TLR-independent pathway, resulting in IFN-β secretion [202]. The modified vaccinia virus Ankara (MVA) vector vaccine activates MyD88 signaling, likely through TLR-9 as well as TLR-independent pathways [203]. Nevertheless, the extent to which TLR and non-TLR innate immune activation contributes to poxvirus-induced adaptive immune responses remains uncertain. A study by Tiegler et al. compared monkeys’ serum cytokines and chemokines levels induced by vaccinations with different poxviruses—ALVAC, NYVAC, and MVA—and demonstrated that each of these vaccines elicited distinct innate immune profiles [204]. This study revealed potentially significant biological distinctions among various poxviruses. Notably, ALVAC (used in RV144) triggered a distinctive proinflammatory cytokine and chemokine reaction upon vaccination in rhesus monkeys and the subsequent infection of human peripheral blood mononuclear cells (PBMCs). However, the extent to which these characteristics confer advantages to vaccines awaits further investigation. Moreover, the effect of innate immune responses in shaping vaccine efficacy has been overlooked.

Indubitably, a sharp paradigm shift that has opened new avenues in HIV vaccinology is the finding that protection is correlated with responses that are non-specific to HIV (e.g., monocytes and NKs), suggesting that balancing the activity of innate and adaptive (virus-specific) responses may be a winning strategy [17]. These findings were corroborated in three independent candidate HIV vaccines (including RV144) in macaques [17,18]. By using systems vaccinology, we reported the activation of both hypoxia and inflammasome pathways within protective monocytes. These data suggest that the RV144-like vaccine in monkeys was effective because it induced monocytes that, in turn, affected adaptive responses via the monocyte/T-cell crosstalk [17,18,205,206]. Unlike conventional vaccines that aim to elicit only specific responses to vaccine-related antigens, inducing long-lasting innate immunity may offer greater protection by stimulating non-specific long-term protective mechanisms (e.g., monocytes/macrophages) that are effective against multiple pathogens and by inducing changes in cells at the epigenetic level. 

Other arms of the innate immune system such as natural killer (NK) and natural killer T (NKT) cells may also act as a bridge between the innate and adaptive immune response to shape the quality and magnitude of the vaccine response. Natural killer (NK) cells may be important component of vaccine immunogenicity, as shown in our RV144 macaque model [17]. Although NK cells are a part of the innate immune system and lack clonal antigen receptors, they are now known to be unique in having adaptive properties of immunologic memory, such as antigen-specific recall responses to a variety of pathogens, most notably to cytomegalovirus (CMV) infection [207,208]. Conclusive evidence of the presence of adaptive memory NK cells was first demonstrated against murine CMV, where NK cells expanded and cleared CMV through a memory-like response [209]. Subsequent studies have affirmed the dominant role of human and rhesus CMV in inducing adaptive memory-like NK cells with epigenetic imprinting, a unique receptor repertoire, and diverse function in humans and macaques [210,211,212,213,214,215]. Additionally, memory-like NK cells without antigen specificity can be induced after cytokine activation with IL-12, IL-15, or IL-18 [216,217]. In recent studies, IL-15 signaling was implicated in the efficacy of RhCMV/SIV vector-based vaccines with protection associated with the magnitude of the IL-15 response and not with the magnitude of the virus-specific CD8+ T cell response [218]. This raises the question of whether NK memory influences trained immunity via monocytes and whether it can be harnessed to improve vaccine efficacy [217,219].

Invariant NKT (iNKT) cells are unique immunomodulatory innate T-cells with an invariant T cell receptor alpha (TCRα) that recognizes the glycolipids presented on Major histocompatibility complex (MHC) class-I-like CD1d molecules. Activated iNKT cells rapidly secrete pro-and anti-inflammatory cytokines, potentiate immunity, and modulate inflammation. Due to their rapid response and broad functional potential, iNKT cells bridge the gap between innate and adaptive immunity [220]. Once activated, iNKT cells can be directly cytolytic (through perforin and granzyme B) and display Th1, Th2 and Th17 effector functions. Additionally, iNKT cells rapidly influence the function of multiple immune subsets. Bidirectional interactions between iNKT and dendritic cells (DC) enhance DC maturation and facilitate antigen cross-presentation and the priming of antigen-specific T-lymphocyte responses; IFNg production by iNKT rapidly activates NK cells, improving cytolysis. iNKT cells are also known to recruit and provide help to B-cells, improving B-cell maturation, antibody class-switching and overall humoral immunity [221,222]. iNKT cell activation was shown to enhance antigen-specific CD4+ and CD8+ T cell responses to a HIV DNA vaccine in mice, with the effect being observed during DNA priming [223]. We demonstrated the effects of iNKT activation in an NHP model of Mauritian-origin cynomolgus macaques and showed the downstream activation effects on CD4+ T-lymphocytes, monocytes, dendritic cells and B cells [224]. Harnessing the immunotherapeutic potential of iNKT activation may be another useful tool for potentiating HIV vaccine efficacy, which can be tested in the NHP model.

***Preexisting immunity effect on HIV vaccines*.** Preexisting immunity to HIV can exert both positive and negative effects on the efficacy of HIV vaccines, encompassing specific antibodies, immune responses, or memory cells. While this immunity may aid in controlling the virus in infected individuals, it can complicate vaccine development. Campion et al. obtained intriguing results in a study conducted as part of the HVTN106 phase I trial, examining the role of cross-reactive memory CD4+ T cells in the primary immune response to HIV-1 gp160 Env. Utilizing ultrasensitive quantification and epitope mapping, the study revealed the presence of both naive and memory CD4+ T cells specific to Env in individuals unexposed to the virus. Surprisingly, the vaccine primarily elicited primary immune responses from the preexisting memory CD4+ T cell pool. This finding underscores the phenomenon of “original antigenic sin” within early vaccine-induced T cell responses, emphasizing the significance of preexisting memory T cells in shaping the immune response to novel pathogens [225,226]. In the realm of HIV vaccine development, it has been noted that even the most potent and broadly neutralizing antibodies, upon being reverted to their inferred germline versions representing naive B cell receptors, frequently exhibit an inability to bind to the HIV envelope. This suggests that the initial B cell response comprises not only naive B cells, but also a pre-existing pool of cross-reactive, antigen-experienced B cells that undergo expansion upon exposure to Env. As part of the HVTN 105 trial, researchers isolated gp120-reactive monoclonal antibodies (mAbs) from participants. Through deep sequencing and lineage tracking, it was discovered that several of these antibody lineages were present in the participants’ pre-immune peripheral blood. Furthermore, these lineages persisted in the post-vaccination bone marrow, particularly within the long-lived plasma cell compartment. Interestingly, the pre-immune lineage members included not only IgM, but also IgG and IgA, and they exhibited somatic hypermutation. These findings suggest that vaccine-induced gp120-specific antibody lineages originate from both naive and cross-reactive memory B cells, underscoring the complex interplay of B cell populations in the immune response to HIV [227].

***BCG and HIV vaccines.*** Preexisting immunity to HIV that arises from other vaccinations or coinfections against/with pathogens is an important area of research in the context of HIV vaccine development [228,229,230,231]. This phenomenon is called “heterologous immunity” or “cross-reactive immunity”, or HIV epitope mimicry [232,233]. Some of the most used vaccines are vaccinia and Bacillus Calmette–Guérin (BCG) for tuberculosis (TB). These vaccines have been very effective in providing protection from the corresponding pathogen and led to the eradication of smallpox. In addition, the nonspecific effect of these vaccines in preventing infections with unrelated pathogens, including HIV, is well documented through epidemiological studies [234,235,236,237,238] (reviewed in [239]). Vaccination with BCG in infants has been demonstrated to decrease mortality caused by childhood infections like respiratory infections and sepsis, which are unrelated to tuberculosis, in high-mortality regions as well as in the USA and Europe [231,240,241,242,243]. Similarly, research conducted in animal models has indicated that immunizing mice with BCG prevents the development of type 1 diabetes (T1D) and confers resistance to vaccinia virus infection [244,245]. Currently, the BCG vaccine is licensed for intradermal delivery in newborns. However, this method does not offer protection against pulmonary disease in adults and cannot be used in individuals living with HIV. Recently, the intravenous administration of BCG has been investigated in adult macaques, including those infected with SIV; [231,246]. Although intravenous BCG administration is not a translatable vaccine in humans, these studies demonstrated favorable safety profiles and contributes to the advancement of understanding correlates of protection in TB.

BCG vaccination has also been investigated as an adjuvant or a vaccine in preclinical settings (Table 1). These approaches aim to leverage BCG’s potential role in enhancing the immune response to HIV. Indeed, BCG is recognized for its nonspecific immunomodulatory effect on the immune system. It can activate various components of the immune system, including innate immune cells, earning it the designation as the gold standard for trained immunity [247,248]. In this context, it has been proposed that the BCG may “train” the myeloid monocyte/macrophage lineage to provide protection against TB and unrelated pathogens [249,250]. Studies have shown that BCG functions as a “self-adjuvanted” vaccine, engaging multiple pattern recognition receptors (PRRs) like Toll-like receptors (TLR2, TLR4, TLR8) to enhance vaccine-induced immunity [251,252]. BCG also induces epigenetic reprogramming in bone marrow myeloid precursors, leading to protection against various unrelated pathogens. In turn, BCG-trained monocytes boost responses to different exposures through non-antigen-specific mechanisms, including increased cytokine and chemokine release and the support of memory adaptive responses [253]. Recent findings also suggest that BCG induces sustained changes in the T cell repertoire, potentially contributing to long-term protection [254]. Applying trained immunity to enhance specific HIV responses across multiple clades is an intriguing concept, with trained monocytes initiating the response; this is followed by effective adaptive cytotoxic responses against conserved HIV regions. BCG’s potential role in HIV vaccine development remains an area of active research. So far, various BCG-based HIV/SIV vaccines have been tested in non-human primates [255,256,257,258]. The results show promising immune responses and are summarized in Table 1. Recombinant (r)BCG (-Tokyo) and Vaccinia Virus (DIs) were tested in combination in cynomolgus macaques [257]. The rBCG expressing full-length Gag was used as a prime, and the non-replicating vaccinia virus was administered as a boost. High-level IFNg responses were detected, and vaccinated monkeys were protected from high viral replication and CD4+ T cell depletion for a year after intrarectal exposure to a pathogenic SHIV clone when compared to the controls. A rBCG and Ad5 combination strategy was tested in rhesus macaques [255]. Strong polyfunctional CD8+ T cells were induced by rBCG-expressing SIV Gag and Pol and rAd5-expressing SIV antigens. The BCG strain AERAS-401 expressing the HIVA immunogen was tested as a prime, followed by vaccination with MVA.HIVA and OAdV.HIVA in RM [256]. The recombinant Mtb strain mc26435 expressing SIV Gag was evaluated for TB- and SIV-specific immune responses in infant macaques. The results showed low levels of SIV-specific immunity following oral and intradermal priming, which was enhanced after the boosts [256,259]. BCG-SIVgag constructs acted as a strong SIV-specific prime for cellular immune responses, inducing SIV-specific CD8+ and CD4+ T-cell responses after the prime. Maintenance of immunogenicity was observed more than 2 years following prime–boost administration, though no protective effect was measured against repeated SIVmac251 rectal mucosal challenge [258].

The studies varied in the animal model, age, vaccine strategy, and readouts used; however, collectively, they demonstrate the potential of mycobacterium-based HIV-1/SIV vaccines to induce specific immune responses in non-human primates. Some approaches show promising results in terms of immunogenicity and protection against viral challenges, while others highlight the need for further strategies to overcome challenges such as immunodominance. Long-term immunogenicity is observed in several cases, but the quest for a fully protective vaccine continues [263].

***CMV and HIV vaccines***. Human CMV (HCMV) impacts almost every part of the host immune system. Studies in identical twins discordant for CMV infection differ in >50% of about 200 immune parameters, providing strong evidence for its influence in shaping the immune landscape [264]. As previously discussed, HCMV profoundly impacts NK cells and is a major driver of NK memory [265]. HIV and CMV appear to work together to exaggerate the effects of CMV on NK cell differentiation [266,267]. Combined with the impact of these differentiated NK cells on early viremia, CMV could modulate vaccine-induced innate and adaptive immune responses. NK cells also play a significant role in the defense against herpesviruses and have a particularly unique relationship with CMV. Indeed, the co-evolution of CMV and the human immune system has led to the expansion of a unique memory-like NK cell subset that is not found in naïve hosts. CMVs are highly species-specific and have co-evolved with their respective host species [268]. The CMV species that is most closely related to HCMV, and that can be experimentally studied, is infection of rhesus macaques with RhCMV [269,270,271,272]. RhCMV is widely prevalent in group-housed captive rhesus macaques in the SPF colony at the Tulane Primate Research Center (TNPRC) and recapitulates many of the known features of HCMV, including natural history and its effect on the immune system [210,270,271,272,273,274]. CMV’s effect on shaping the immune system could therefore have consequences on the host response to vaccines, including preclinical AIDS vaccine testing. There are conflicting data on the effect of CMV co-infection on vaccination [219]. One study reported both higher and lower anti-influenza antibody responses depending on age [270], and another observed increased influenza vaccination-induced antibody responses in CMV+ compared to CMV− macaques [275]. Because of the effect of CMV seropositivity on alterations in the T cell repertoire and immunosenescence, its impact on vaccine responses remains an important, albeit unresolved, consideration [276,277].

## 6. Future Directions

Extensive preclinical and clinical testing has highlighted that targeting one facet of the immune system may not be an effective strategy for achieving a globally effective HIV vaccine. The lessons learned from the T cell vaccine era (second wave) emphasize that adopting a “more is better” approach, such as aiming for strong CD8+ T cell responses or higher levels of interferon-gamma (IFNg) as an efficacy marker, has proven inadequate. Intriguingly, the only vaccine to achieve partial protection against HIV acquisition did not induce strong anti-viral CD8+ T cells, but protective CD4 and non-neutralizing antibodies to the V2 loop. Furthermore, this same vaccine failed to reduce the HIV replication levels in individuals who became infected. It appears that safeguarding against viral replication and acquisition may require distinct immune responses. This notion finds support in the observation that successful viral load control in macaques did not translate into protection against acquisition in humans, and in some instances, it even heightened the risk of viral acquisition. It is therefore worthwhile to reconsider the strategy by emphasizing a short-term vaccine approach focused on preventing acquisition rather than striving for long-lasting protection (like bNabs approaches). Alternatively, new adjuvants could be developed to circumvent the necessity of inducing CD4+ T cells or to redirect CD4+ T cells away from transmission sites, possibly using innovative strategies, such as chemokines, heat shock proteins, or other delivery strategies aimed at inducing site-specific responses [278,279].

We find ourselves in what could be considered a new era of vaccine development, although it can be challenging to accurately define waves while actively experiencing them. Undoubtedly, the emergence of the COVID-19 pandemic has transformed how we disseminate scientific information and, most importantly, has underscored the urgency of expediting vaccine testing in human subjects. Current challenges in the preclinical field include the availability of specific NHP models in the post-COVID era and cost constraints. National Primate Centers are actively expanding their colonies due to disruptions in supplies from other countries. Cost limitations are particularly pertinent in low-dose repeated challenge models, where achieving statistical power necessitates larger animal groups. Furthermore, conducting side-by-side comparisons of multiple vaccines can be financially burdensome and often impractical unless clear indicators of protection have been identified and thoroughly studied. Despite these considerations and challenges, the NHP model remains an indispensable tool in vaccine research, offering a crucial bridge between preclinical studies and the complex human clinical trial phase.

Finally, by examining the ongoing efforts and advancements in HIV prevention research, we hope to contribute to the collective knowledge and foster new ideas that can pave the way for the development of an effective HIV vaccine.

## Figures and Tables

**Table 1 viruses-16-00368-t001:** Summary of in vivo studies on BCG-based HIV-1/SIV vaccines in non-human primates.

Vaccine	Animal	Result	Reference
rBCG (full-length SIV Gag) + Vaccinia virus boost	Cynomolgus Macaques	High IFNg secretion, protection from viral challenge, observed for a year; no protection with separate vaccine modalities	[257]
rBCG (SIV Gag and Pol) + rAd5 boost	Rhesus Macaques	Induced polyfunctional CD8+ T-cell profile	[255]
AERAS-401 prime + MVA.HIVA and OAdV.HIVA boost	Rhesus Macaques	High-frequency HIV-1-specific T-cell responses; safety demonstrated; lower T-cell immunogenicity in infants	[260]
rMtb mc26435 expressing SIV Gag + MVA boost	Infant Macaque Model	Low levels of SIV-specific immunity, enhanced after boosts	[256]
		Mucosal SIV-specific IgA in saliva and intestinal IgA and IgG	[259]
rBCGpan-Gag prime + Gag VLP boost	Chacma Baboons	Gag-specific responses after two primes, enhanced by Gag VLP boost	[261]
Minigenes + rBCG, rDNA, rYF17D, rAd5 combinations	Rhesus Macaques (Mamu-A*01 + MHC-1)	Modest reduction in viral set point following SIVmac239 challenge; need for strategies to overcome immunodominance	[262]
rBCG-SIVgag constructs	Rhesus Macaques	strong SIV-specific prime for cellular immune responses; maintenance of immunogenicity over 2 years; no protective effect	[258]

## Data Availability

No new data were created.

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
