# Peer review of "Exploring HIV Vaccine Progress in the Pre-Clinical and Clinical Setting: From History to Future Prospects"

_viruses, 2024, doi:10.3390/v16030368_

Round 1

Reviewer 1 Report

Comments and Suggestions for Authors

The manuscript submitted by Kaur and Vaccari provides a  thorough review of the clinical and preclinical development of HIV vaccines. As detailed in the English language section, a few areas of the manuscript could benefit from editing. The manuscript would also benefit from adding more scientific details or alternative phrasing. 

On lines 46-47, the authors state that it was found that CD8 T-cell responses alone will not eliminate the virus. This is too definitive. This section only refers to specific CD8 T-cells induced by the discussed vaccines. Indeed, later in the manuscript, the authors mention CMV-based vaccines that induce CD8 T-cells that might clear infection in NHPs. 

At various points throughout the manuscript (e.g., lines 67-69 and 235-237), the authors state that passive provision and active immunization are strategies to "elicit" bNAbs by vaccination. Discussing these as strategies to "establish bNAbs in vivo" would be more appropriate, as passive provision does not elicit bNAbs. 

It is incorrect to say RV144 did not induce neutralizing antibodies or CD8 T-cells. Neutralizing antibodies to tier 1 viruses and anti-viral CD8 T-cells were induced in a proportion of participants. 

Line 199 - HVTN 097 differed from RV144 regarding adjuvant and viral subtype. How can the formulation be the same?

Lines 252-253 - what type of neutralizing antibodies were generated by the early gp120 protein vaccines? Were these tier-1 neutralizing Abs? I think this should be clarified. 

Line 257 - It would benefit the reader for the authors to elaborate on what encouraging results were observed.

Line 291 - A reference formatting issue is present for Klatt 2012. 

Lines 333-335 - SHIVs do not overcome the problem of distinct CTL epitopes compared to HIV, as most SHIVs retain SIV proteins, such as gag.

Line 349 - It is incorrect to say that the IV challenge model has no clinical relevance. Some populations contract HIV by sharing drug injection equipment.

Efforts to generate and utilize NHP models of SIV or SHIV cell-associated virus challenge should be discussed. 

Lines 469-471 - How well does SF162P3 reflect the HIV variants people are exposed to? Is there a problem with NHP exposure models per se, or is there a problem with the viruses used for challenges?

Line 484 - "similar to humans" - the original RV144 study did not assess mucosal responses.

Section 5: Innate immunity - It might be beneficial to address the fact that HIV and CMV appear to work together to exaggerate the effects of CMV on NK cell differentiation (PMID: 36053502; PMID: 27314055). Combined with the impact of these differentiated NK cells on early viremia, these data could provide insight into the role of these cells in immunization efforts. 

Table 1 - The study and animal columns are identical. The table should be reformatted. 

The paper would benefit from a figure or two to summarize information.

Lines 714-716 - Could one argue that passive immunization with bNAbs is an exception to the statement that targeting one facet of the immune system is insufficient to achieve protection? Particularly given that some bNAbs (e.g., PGT121) appear to protect with Fc-dependent effector functions.

Line 730 - elaborate on "other strategies".

Comments on the Quality of English Language

Lines 16-18 - "... and it plays..." revision needed to keep the sentence consistent with the plural nature of "NHP models". Revise to "... and they play... before they can proceed ...".

Line 23 - The sentence starting with "progress" needs to be revised to add a capitalized "P".

The sentence spanning lines 47-50 should be revised for clarity. 

Line 156 - insert "and" between "safety" and "showed".

Line 237 - delete "assessing".

Line 411 - "... it was found..." - what is "it"? 

Author Response

Thank you for your valuable feedback. We have carefully reviewed the comments and suggestions provided by the reviewer and made the necessary revisions to the manuscript. The changes have been highlighted in red for easy identification. Below is a point-by-point response addressing each comment:

  1. We have edited lines 46-48 to reference the vaccines cited at line 44 as requested.

  2. The section related to establishing bNabs has been expanded and corrected (see lines 79-88).

  3. Corrections have been made regarding the RV144 results, and literature has been added to support the paucity of some CD8 T cell responses in RV144 (see lines 264-275 and 291-298).

  4. Additional information has been included to differentiate between HVTN 097 and RV144, with clarification regarding changes in protein adjuvant and d clade introduced in HVTN 100 (see lines 326-335).

  5. The discussion on neutralization sensitivity tiers has been expanded (see paragraph 377-383).

  6. The reference has been corrected.

  7. The correction for line 333 is now reflected in lines 518-518.

  8. The correction for line 345 is now shown in lines 535-538.

  9. Lines 469-471 have been updated to 664-666, including commentary on SF162P3 as the challenge virus.

  10. References have been added in the innate paragraph.

  11. The table has been corrected.

  12. Line 730 is now indicated as line 961.

Thank you for your thorough review, and we believe these changes significantly enhance the clarity and accuracy of the manuscript.

Reviewer 2 Report

Comments and Suggestions for Authors

In this review, Kaur and Vaccari provide an overview of the current state of preclinical and clinical based HIV vaccines and some potential new avenues of exploration. The manuscript is overall well written and covers a variety of topics that would be of interest to the readership. At times the level of detail of studies covered is not deep enough and this is likely because of the breadth of studies being examined. As the major emphasis of this review is on using NHP studies to study HIV/SIV vaccines, the manuscript would benefit from utilizing the expertise of the authors by going into further detail into the NHP studies and to give a more generalized overview of HIV clinical development, which have been described more thoroughly reviewed elsewhere.

Major

1.     Table 1 introduces cynomologous macaques, baboons, an infant RM as models for testing HIV/SIV vaccines however these models are discussed in earlier sections. Thus, it is unclear what the pros/cons are of these models relative to more traditionally define ones.

2.     As written, the section on “CMV natural infection in HIV vaccines” seems a bit out of place with the rest of the review. A bigger emphasis on how CMV is known to impact HIV/vaccines or more generally how pre-existing immunity to co-infections shapes the immune landscape during HIV/SIV vaccination would tie more effectively to the rest of the manuscript.

Minor

1.     The “Recent HIV Vaccine Development” section would be strengthened by appropriate citations.

2.     Could the authors comment on how circulating clades of HIV in the different regions where human vaccine trials occurred may have impacted the efficacy of the vaccines.

3.     Acronyms should be defined during their first use, this is inconsistently or not done throughout the manuscript. Acronyms are also sometimes inconsistently used e.g. “i.vag” vs “Ivag”.

4.     A sentence or two describing the tested vaccine constructs (i.e. type of vaccine, vaccine targets, etc) should be added to provide context to the reader.

5.     Line 220 “Considerations of the many distinctions between the South African and Thai trials are important to prevent any mistaken inference that the results of the former trial undermine those of the latter,” please elaborate on what these considerations are.

6.     Section 3-5 may be more appropriately placed prior to Section 2 as they provide an overview of the preclinical studies that provide the framework for the described human clinical studies.

7.     This reviewer would be wary in using strong language that devalues the use of NHP preclinical HIV vaccine studies, as these phrases could be taken out of context e.g. Line 395 “reliability”, line 408 “significant disparity”, line 448 “relevance.” Re-wording to instead highlight weakness of different NHP models is recommended.

8.     Line 411, what is the specific vaccine that is being referenced to.

9.     Lines 487-189. Is the author’s intention to say that some NHP vaccine studies better predicted the outcomes of human clinical trials, please clarify.

10.  It would be worth discussing and including the reference DOI: 10.1038/s41591-021-01574-5 to the manuscript to the discussion of mRNA HIV vaccines and the reference DOI: 10.1371/journal.ppat.1009278 to the discussion on innate immunity

Comments on the Quality of English Language

The manuscript is generally written well, however there are multiple typos particularly in the second half of the manuscript that reduce the readability or interpretation at times, which may need to be addressed during the revision or during editing.

Author Response

Thank you for your valuable feedback. We have carefully reviewed the comments and suggestions provided by the reviewer and made the necessary revisions to the manuscript. The changes have been highlighted in red for easy identification. We have increased the details on the rationale and results of studies covered within the overall review. We hope that we have now struck a better balance between the breadth of the topics and the depth of discussion. Below is a point-by-point response addressing each comment:

I) We acknowledge the point regarding the variables in the studies however we think that model considerations may not be informative in this context given the many other variables between these studies. We added a comment at Line 911.

II)The CMV section was revised accordingly.

1)We have corrected the citations.

1-4)Additional information on Clades has been included throughout the text (see lines 11, 121, 255, and 326). We have also corrected acronyms and provided more details on vaccine constructs throughout the review.     

5) Line 220 is now line 354, and we have elaborated on the differences between RV144 and HVTN 100 in lines 356-360.

6)A paragraph has been added for a better transition between human and macaque studies (see lines 429-435).

7-9). Corrections have been made accordingly, and clarifications have been added.

10) References have been included (see lines 730-732).

We believe these revisions significantly enhance the clarity and accuracy of the manuscript. 

Reviewer 3 Report

Comments and Suggestions for Authors

In Kaur and Vaccari, the authors have provided a succinct evaluation of the HIV vaccine field and its history.  The authors describe previous clinical trials of various HIV vaccine strategies and then discuss NHP models for how they have been used to evaluate new candidate vaccines.  This is especially important because due to the cost and ethical considerations, clinical trails for every vaccine strategy are not feasible.  The authors touch on key topics for the NHP model like macaque species, challenge virus, challenge route, dose and ultimately, what correlates are important should protection be observed.

This is a nice review to help summarize the HIV vaccine history as of 2024.  There are only a few comments that could help improve this manuscript:

Line 124 – Was the Ad26 mosaic vaccine thought to be able to induce neutralizing antibodies?  I thought the idea was attempt to generate some antibodies to help the T cell responses.  Please clarify.

Line 139 – Please include the confidence interval for the 25% efficacy statement.  This would give the reader a better picture of the results.

Line 151 – It would be helpful to include the component details of ALVAC-HIV and AIDSVAX and some context for why they were combined.  Were they ever tested individually?

Line 226 – Considering this is a review, it might be interesting to provide some commentary on the follow-up trials of RV144 failed in the Africa.  What were key differences or limitations in the new trials?  How does this reflect on the results of RV144?

Line 275 – There needs to be a better transition going from the SOSIP topic to NHP models.

Line 291 – Reference format error for Ref #61.

General comment – It would be helpful to include some of the early SIV vaccine work in NHPs in the 80’s and 90’s.  Specifically the studies where anti-MHC antibodies were protective against SIV challenge because the vaccine and challenge virus were made from the same cell line and the deltaNef vaccines.  It would be a good history lesson on how to assess vaccines in NHP models.

Author Response

Thank you for your valuable feedback. We have carefully reviewed the comments and suggestions provided by the reviewer and made the necessary revisions to the manuscript. The changes have been highlighted in red for easy identification. As suggested in the general comment section, we have added historical information, including on early SIV vaccines in NHP (lines 458-479). Below is a point-by-point response addressing each comment:

  1. To our knowledge, the mosaic strategies were designed to induce protective cellular responses against heterologous circulating strains. We realized that we did not do a good job in differentiating mosaic strategy from the MOSAICO trial, and we have now expanded on these concepts for clarity (and added history) in the clinical trials with Adenovirus vaccines section.

  2. We have included the confidence interval.

  3. Details have been included in all paragraphs.

  4. We added commentary on differences (lines 356-361).

  5. We agreed and added a transition.

  6. We corrected the reference format.

We believe these revisions significantly enhance the clarity and accuracy of the manuscript. 

Round 2

Reviewer 2 Report

Comments and Suggestions for Authors

The authors have adequately addressed all concerns raised during the initial review. No additional concerns are noted.

Comments on the Quality of English Language

General editing may be required during the proof stage.